# A Program to Reduce Post-Operative Opioid Prescribing at a Veteran’s Affairs Hospital

**DOI:** 10.3390/jcm11185453

**Published:** 2022-09-16

**Authors:** Callie Hlavin, Sruthi Muluk, Visala Muluk, John Ryan, Jeffrey Wagner, Rajeev Dhupar

**Affiliations:** 1Department of Surgery, University of Pittsburgh, Pittsburgh, PA 15213, USA; 2School of Medicine, University of Pittsburgh, Pittsburgh, PA 15213, USA; 3Department of Medicine, Veteran’s Affairs Pittsburgh Healthcare System, Pittsburgh, PA 15240, USA; 4Department of Cardiothoracic Surgery, University of Pittsburgh, Pittsburgh, PA 15213, USA; 5Division of Clinical Pharmacy, Veteran’s Affairs Pittsburgh Healthcare System, Pittsburgh, PA 15240, USA

**Keywords:** opioid deprescribing, acute pain, surgery, post operative, Veteran’s affairs

## Abstract

Variability in surgeon prescribing patterns is common in the post-operative period and can be the nidus for dependence and addiction. This project aims to reduce opioid overprescribing at the Veteran’s Affairs Pittsburgh Healthcare System (VAPHS). The VAPHS Opioid Stewardship Committee collaborated to create prescribing guidelines for inpatient and outpatient general, thoracic, and vascular surgery procedures. We incorporated bundled order sets into the provider workflow in the electronic medical system and performed a retrospective cohort study comparing opioid prescription patterns for Veterans who underwent any surgical procedure for a three-month period pre- and post- guideline implementation. After implementation of opioid prescribing guidelines, morphine milligram equivalents (MME), quantity of pills prescribed, and days prescribed were statistically significantly reduced for procedures with associated guidelines, including cholecystectomy (MME 140.8 vs. 57.5, *p* = 0.002; quantity 18.8 vs. 8, *p* = 0.002; days 5.1 vs. 2.8, *p* = 0.021), inguinal hernia repair (MME 129.9 vs. 45.3, *p* = 0.002; quantity 17.3 vs. 6.1, *p* = 0.002; days 5.0 vs. 2.4, *p* = 0.002), and umbilical hernia repair (MME 128.8 vs. 53.8, *p* = 0.002; quantity 17.1 vs. 7.8, *p* = 0.002; days 5.1 vs. 2.5, *p* = 0.022). Procedures without associated recommendations also preceded a decrease in overall opioid prescribing. Post-operative opioid prescribing guidelines can steer clinicians toward more conscientious opioid disbursement. There may also be reductions in prescribing opioids for procedures without guidelines as an indirect effect of practice change.

## 1. Introduction

Drug overdose is the leading cause of injury-related death in the United States with over two-thirds of the 700,000 deaths between 1999 and 2007 involving prescribed opioids [1,2]. In 2017, Pennsylvania ranked the third highest in age-adjusted overdose death rates [3]. Overprescribing and wide variations in prescribing patterns associated with the nidus for dependence and addiction are major contributors to the current epidemic [4,5,6,7,8]. Further examination indicates that variations in prescribing patterns are particularly common in the post-operative period.

Current studies show that up to 75% of post-operative patients have unused opioid pills after discharge [9,10]. Although inadequate pain control can lead to decreased mobility and higher readmission rates, this excess indicates that current prescribing patterns overcompensate for pain control. Providers not tailoring opioid prescriptions to procedure or inpatient opioid use may contribute to this overcompensation. Locally, in Pittsburgh, our group has studied this phenomenon and found that most patients were over-prescribed opioids on discharge after thoracic surgery [11].

In this growing dilemma, the Veteran Affairs (VA) population had increasing concerns about opioid use for chronic pain. In response, the VA implemented the Opioid Safety Initiative in 2013, which disseminates best practices for prescribers, patient overdose prevention education, and prescription tracking. This initiative led to decreased opioid prescriptions after total knee arthroplasty [12]. To further target the opioid crisis at its inception, this study aims to decrease the number of opioids in circulation by guiding clinicians to adopt more thoughtful post-operative prescribing practices through the creation and implementation of multidisciplinary, clinician tailored, and patient-centered post-operative opioid prescription order sets. The goal of these post-operative prescribing guidelines is to reduce opioid dispensing volume in the Pittsburgh Veteran’s Affairs Health System (VAPHS).

## 2. Materials and Methods

The VAPHS Opioid Stewardship Committee created post-operative prescribing recommendations for inpatient general, vascular, and thoracic surgery procedures and were created based on the patient’s opioid requirement in the 24 h prior to discharge (Table A1). We created outpatient general, vascular, and thoracic surgery procedure guidelines by calculating the average number of opioids prescribed per procedure at the VA and reducing this by 75% (Table A2, Table A3 and Table A4). These recommendations were discussed, modified, and approved by respective department chiefs. Guidelines were created for patients undergoing both inpatient and common outpatient procedures. Bundled order sets were created and incorporated into the provider workflow in CPRS—the VA electronic medical system from these guidelines (Table A1, Table A2, Table A3 and Table A4). In addition to the order set, 30-min annual education sessions for the prescribers were implemented to demonstrate the order set and discuss the rationale.

A retrospective, observational study investigating the change in post-operative opioid prescribing after guideline implementation was conducted according to STROBE guidelines (Table A5). Quantity of opioid pills and number of days of the prescription were collected by the pharmacy department for all procedures over three-month intervals both pre- and post-guideline implementation. Because nearly all Veterans fill their prescriptions at the VA pharmacy, we matched patients and prescriptions by date of surgery and patient identifiers, including name and date of birth. The total morphine milligram equivalents (MME) for each prescription were calculated based on quantity of opioid pills and number of days. The pre-implementation period was between August and November of 2019 and the post-implementation period was between July and October of 2020. The authors selected the eight most performed procedures for comparison—four with associated post-operative opioid prescribing guidelines and four without associated guidelines. The four procedures with associated guidelines were laparoscopic cholecystectomy, inguinal hernia repair, umbilical hernia repair, and video-assisted thoracoscopic surgery (VATS). The four procedures without associated guidelines were kidney transplant, carpal tunnel release, laminectomy, and coronary artery bypass graft (CABG).

Data was analyzed separately for each procedure using negative binomial regression with the quantity of opioid pills, number of days of the prescription, and MME regressed on the timepoint (pre/post). Estimated marginal means were then extracted from the statistical model for the pre- and post-timepoints. Data were analyzed in R (v. 4.1.2), and a *p* < 0.05 after Benjamini-Hochberg multiple test-correction was considered statistically significant [13].

## 3. Results

After implementation of opioid prescribing guidelines, most procedures with associated post-operative opioid prescribing guidelines preceded a decrease in opioid prescribing. Morphine milligram equivalents (MME), quantity of pills prescribed, and days prescribed were statistically significantly reduced for cholecystectomy (MME 140.8 vs. 57.5, *p* = 0.002; quantity 18.8 vs. 8, *p* = 0.002; number of days 5.1 vs. 2.8, *p* = 0.021), inguinal hernia repair (MME 129.9 vs. 45.3, *p* = 0.002; quantity 17.3 vs. 6.1, *p* = 0.002; number of days 5.0 vs. 2.4, *p* = 0.002); and umbilical hernia repair (MME 128.8 vs. 53.8, *p* = 0.002; quantity 17.1 vs. 7.8, *p* = 0.002; number of days 5.1 vs. 2.5, *p* = 0.022). Video-assisted thoracoscopic surgery (VATS) found a significant decrease in MME (113.4 vs. 67.5, *p* = 0.045) (Table 1).

Procedures without associated post-operative opioid prescribing guidelines also preceded changes in opioid prescribing. Coronary artery bypass grafting (CABG) preceded a statistically significant increase in number of days of opioids prescribed over pre- and post- time intervals (9.55 vs. 12.64, *p* = 0.049). Over pre- and post- time periods, laminectomy (MME 530 vs. 266, *p* = 0.002; number of days 22.93 vs. 9.86, *p* = 0.002; quantity 70.7 vs. 28, *p* = 0.002) and carpal tunnel release (MME 155 vs. 122, *p* = 0.0037; number of days 6.67 vs. 4, *p* = 0.002; quantity 21 vs. 16.5, *p* = 0.002) had significant decreases in all opioid prescribing measures. Kidney transplants did not have a statistically significant change in opioid prescribing (Table 1).

Including all patients, procedures with associated guidelines had a significant reduction in MME and days, but not quantity. Procedures without associated recommendations also preceded a decrease in overall opioid prescribing.

## 4. Discussion

In this study, we found that creation and implementation of post-operative prescribing guidelines can successfully influence clinicians towards more responsible opioid distribution in a Veteran’s Affairs hospital setting.

Multiple successful institutional opioid prescribing guidelines have been published and utilize tools such as prescriber education, enhanced oversight, and prescription calculators. Coinciding results from implementation of these guidelines are ubiquitous throughout multiple surgical subspecialties. Lee et al., investigated the effect of education and prescribing guidelines for breast and melanoma procedures and found that the number of opioids prescribed decreased without a compensatory increase in refills [14]. Additionally, tracking prescriptions in ophthalmologic surgery found that both the frequency and mean oral morphine equivalent decreased after guideline distribution, but the number of refill prescriptions remained unchanged [15]. State-mandated restrictive policies seem to be effective as well [16,17,18,19]. In Vermont, the Department of Health issued new rules governing the prescribing of opioids for pain in July 2017 and, consequently, the University of Vermont Medical Center saw a decline in the median morphine milligram equivalents (MME) prescribed for fifteen operations across four surgical specialties without any increase in refill rate [20]. Guidelines are successful in reducing opioid prescriptions while not lowering patient satisfaction [21,22,23]. Widely renowned medical centers such as the Mayo Clinic and John’s Hopkins as well as state-wide campaigns, such as in Michigan and Illinois, have recognized the severity of the opioid epidemic and the key role of hospital providers [16,17,18,19]. In response, there have been many versions of opioid guidelines; however, none have targeted the Veteran population.

In the current study, the creation of guidelines was based on the current literature related to opioid prescribing, which suggests that 70–80% of post-operative opioids are unused and that inpatient opioid use in the 24 h prior to hospital discharge reflects opioid need after discharge [9,24]. The authors adopted a multidisciplinary approach for the creation of post operative opioid prescribing guidelines. This method involved surgeons, medical physicians, and pharmacists in conjunction with hospital leadership. The researchers involved department chiefs in the revision and approval of drafted opioid prescribing guidelines. The incorporation of a diverse group of collaborators encouraged hospital staff to readily accept guideline adaptations.

Prescribing guidelines have shown success in state-wide campaigns and publicly funded hospital centers. This study documents the first post-operative opioid prescribing guidelines in a Veteran surgical population. When informally queried, providers who used the guidelines reported that they aided in provider workflow and made prescribing a more efficient process. Additional benefit was seen with intern and resident physicians who did not have experience with opioid prescribing. Ultimately, guidelines assisted naïve providers in prescribing appropriate number and days of opioids.

Results from this study suggest that creation and implementation of post-operative opioid prescribing guidelines with education helps to reduce opioid prescribing to more appropriate levels. Across general and thoracic surgery procedures with associated guidelines, nearly all measures of opioid prescribing preceded a statistically significant reduction after guideline implementation (Table 1). Additionally, there may also be decreases in opioid prescribing for procedures without guidelines as an indirect effect of practice and culture change.

Although this study provides evidence of opioid prescribing guideline benefits, it is not without limitations. The post-implementation data collection period occurred during the COVID-19 pandemic, which may alter results due to a change in the number of elective procedures. However, we did not observe a consistent difference in the size of the post-implementation patient samples. Additionally, our pre- and post- implementation collection periods were relatively short, thus leading to small sample sizes which limits generalizability of the results. We collected opioid prescription data from phamarcy records and this data was then manually sorted. As such, there were time and personnel limitations to the amount of data collected and a formal sample size calculation was not completed. The short collection periods were intended to serve as “proof of concept” with more robust data collection to follow in future studies. The opioid guidelines targeted opioid naïve patients, yet data collection did not exclude chronic opioid users. Lastly, there was no comparison of the pre- and post-implementation patient groups to assess for difference in discharge opioid use, pain management, and satisfaction. The authors anticipated that prescribing guidelines would steer clinicians into more appropriate post-operative opioids prescribing. Formal investigation into improved provider prescribing awareness was beyond the scope of this study but may be useful in further studies. Future research is needed to confirm acceptable patient analgesia after opioid prescribing guideline implementation as well as the effect of these guidelines on long-term prescribing.

## Figures and Tables

**Table 1 jcm-11-05453-t001:** Pre- and post- guideline opioid prescribing in procedures with and without associated guidelines.

	Pre-Guidelines Mean	95% CI	Post-Guidelines Mean	95% CI	*p*-Value	Adj *p*-Value *
Associated with guidelines						
Laparoscopic cholecystectomy (pre: *N* = 9; post: *N* = 10)						
Quantity	18.8	15.03–23.5	8	6.12–10.5	<0.001	0.002
# of days	5.11	3.83–6.82	2.8	1.93–4.06	0.012	0.021
MME	140.8	110.6–179.4	57.5	45.3–72.9	<0.001	0.002
Inguinal hernia repair (pre: *N* = 25; post: *N* = 23)						
Quantity	17.32	15.8–19.03	6.13	5.2–7.23	<0.001	0.002
# of days	5	4.20–5.96	2.43	1.87–3.16	<0.001	0.002
MME	129.9	118.4–142.5	45.3	40.7–50.5	<0.001	0.002
Umbilical hernia repair (pre: *N* = 12; post: *N* = 6)						
Quantity	17.17	14.66–20.1	7.83	5.76–10.6	<0.001	0.002
# of days	5.08	3.96–6.53	2.5	1.51–4.15	0.014	0.022
MME	128.8	106.2–156.1	53.8	40.4–71.5	<0.001	0.002
VATS (pre: *N* = 8; post: *N* = 5)						
Quantity	18.8	13.19–26.6	10	6.17–16.2	0.039	0.052
# of days	5.25	3.78–7.29	3.4	2.06–5.62	0.16	0.19
MME	113.4	85.2–151.1	67.5	46.7–97.6	0.03	0.045
Not associated with guidelines						
CABG (pre: *N* = 20; post: *N* = 14)						
Quantity	38.2	35.1–41.6	35.4	31.9–39.3	0.28	0.31
# of days	9.55	8.01–11.4	12.64	10.43–15.3	0.035	0.049
MME	286	261–314	266	238–297	0.3	0.313
Laminectomy (pre: *N* = 15; post: *N* = 14)						
Quantity	70.7	53.7–93.0	28	15.2–51.7	0.001	0.002
# of days	22.93	18.62–28.3	9.86	7.68–12.7	<0.001	0.002
MME	530	404–696	266	200–353	0.001	0.002
Kidney transplant (pre: *N* = 12; post: *N* = 5)						
Quantity	47.1	31.9–69.5	28	15.2–51.7	0.16	0.19
# of days	12.1	9.51–15.4	8.8	5.89–13.2	0.18	0.21
MME	212	170–263	210	150–294	0.97	0.97
Carpal tunnel release (pre: *N* = 48; post: *N* = 41)						
Quantity	21	19.2–23.0	16.5	14.8–18.3	0.001	0.002
# of days	6.67	5.59–7.95	4	3.23–4.95	<0.001	0.002
MME	155	140–172	122	109–137	0.002	0.0037

* adjusted *p*-value, FDR corrected.

## Data Availability

Data available on request due to restrictions.

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
