# Peer review of "A Program to Reduce Post-Operative Opioid Prescribing at a Veteran’s Affairs Hospital"

_jcm, 2022, doi:10.3390/jcm11185453_

Round 1

Reviewer 1 Report

The authors need to elaborate on the aims in the introduction section.

The last line of introduction states the results, if it describes the current study, it has to be deleted.

Under the methods section, it may be worth describing which approach was used or suggested as to prescribe discharge opioids; either the usage the day prior, figure A1 or procedures A2-A4, On what basis the recommendations were followed?

Authors need to comment on why a formal sample size calculation was not attempted, this is a significant limitation  to the study as the numbers are too small to to validate and generalise the results 

Conclusion is missing.

Reference no 24 is incomplete.

Reviewer 2 Report

The manuscript titled “A Program to Reduce Post Operative Opioid Prescribing at a Veteran’s Affairs Hospital” aimed at creating and implementing post-operative prescribing guidelines, in order to decrease opioid over-prescription after different types of surgery. It focuses on a timely topic of clinical importance, which could help to counteract the epidemic phenomenon of overdose deaths. 

The work is interesting but some action is needed:

- Sample size is very small for each type of surgery. The authors should consider to expand the collection period in order to enhance the robustness of the research results.

- In the abstract, please provide the full form of ‘MME’.

- In the abstract and in the main text, ‘Post operative’ is written either with or without the hyphen. Please review.

- The format of some references does not meet the journal guidelines. Please review.

- Ref 24 should be completed.
